# Temporal trends in the incidence and case severity of COVID-19 cases among the Syrian refugees in Azraq camp in Jordan: A retrospective observational study

Ahmad Waleed Zghool[1], Ahmad Alrawashdeh[2]*, Zaid I. Alkhatib[2], Sara A. Nasser[3]; Natalya Kostandova[4], Shiromi M. Perera[4], Jomana W. Alsulaiman[5], Adi H. Khassawneh[1], Abdel-Hameed W. Al-Mistarehi[6], Amer Abu-Shanab[7], Khalid A. Kheirallah[1]

**1** Department of Public Health and Family Medicine, Jordan University of Science and technology, Irbid, Jordan, **2** Department of Allied Medical Sciences, Faculty of Applied Medical Sciences, Jordan University of Science and Technology, Irbid, Jordan, **3** Department of Clinical Pharmacy, Faculty of Pharmacy, Jordan University of Science and Technology, Irbid, Jordan, **4** International Medical Corps, Washington, District of Columbia, United States of America, **5** Department of Pediatrics, Faculty of Medicine, Yarmouk University, Irbid, Jordan, **6** School of Medicine, Johns Hopkins Hospital, Baltimore, Maryland, United States of America, **7** International Medical Corps, Jordan Country Office, Amman, Jordan

* aaalrawashdeh@just.edu.jo

## Abstract

### Background

Azraq Syrian refugee camp, located in Jordan, is where the challenges of managing the COVID-19 epidemic meet the vulnerabilities of displaced people. This study aimed to investigate the epidemiological characteristics, incidence, risk factors, and outcomes of COVID-19 among Azraq camp residents.

### Methods

COVID-19 data from Azraq camp were collected by International Medical Corps clinics and analyzed retrospectively from August 1, 2020, to August 31, 2022. Data included demographics, risk factors, testing history, contact tracing, and vaccination profiles. We estimated COVID-19 incidence and analyzed risk factors using Poisson and multilevel logistic regression.

### Results

A total of 2,468 confirmed COVID-19 cases were identified, with a prevalence of 5.6 per 100 residents. The camp's monthly incidence rate was more than 50% lower than the national rate, with a 1.7% monthly decrease. Females had a higher incidence than males (6.4% vs. 4.9%, p < 0.001), while the elderly bore the greatest disease burden. Home-based isolation was the main strategy, except during the second wave. Vaccination coverage reached 31.6%, primarily with Pfizer (49.8%). Symptomatic cases made up 44.0% of confirmed cases, with 10.4% requiring hospitalization. Factors independently associated with hospitalization included age, comorbidity, and vaccination status.

**Data availability statement:** Data cannot be shared publicly because of restrictions from the UNHCR. Data are available from the International Medical Corps for researchers who meet the criteria for access to confidential data. Data can be requested through contacting International Medical Corps, DC Office, 2000 M Street NW, Washington, DC 20036, research@internationalmedicalcorps.org.

**Funding:** The author(s) received no specific funding for this work.;

**Competing interests:** The authors have declared that no competing interests exist.

## Conclusion

The study highlights the need for robust surveillance, targeted healthcare interventions, equitable resource allocation, and vaccination campaigns to manage COVID-19 and future epidemics in refugee camps.

## Author summary

The Azraq Syrian refugee camp in Jordan is home to a vulnerable population living under challenging conditions, making it difficult to manage the COVID-19 pandemic effectively. Our study examined the spread and impact of COVID-19 within this camp from August 2020 to August 2022. We collected and analyzed data on the residents' demographics, health risks, testing, and vaccination status. We found that COVID-19 spread less rapidly in the camp than in the general Jordanian population, with the monthly infection rate decreasing over time. Women and older residents were particularly affected, with the elderly experiencing the most severe cases. While most patients were treated at home, some required hospitalization, especially those who were older or had underlying health conditions. Vaccination played a crucial role in managing the pandemic, with nearly one-third of the camp's population receiving the vaccine. Our findings emphasize the need for ongoing health monitoring, tailored medical interventions, and increased vaccination efforts to protect vulnerable communities like those in refugee camps from current and future health crises.

## Introduction

The emergence of severe acute respiratory syndrome coronavirus 2 (SARS-CoV-2) in December 2019 and its rapid evolution into a global pandemic by March 2020, as declared by the World Health Organization (WHO), has emphasized the importance of epidemiological investigation across all sectors of society [1]. The spread of COVID-19 has also challenged health systems, necessitating an in-depth examination of its epidemiology, especially among vulnerable population sub-groups. Among these sub-groups are refugees, whose living conditions are considered suboptimal: overcrowding, limited WASH facilities and limited healthcare access, and social inequities. While these conditions exacerbate susceptibility to infectious diseases and affect disease epidemiology [2], limited research investigated COVID-19 among refugees.

In Jordan, a host country to a considerable number of Syrian refugees, the pandemic's management within refugee settings like Zaatari and Azraq camps has been a critical concern. As the two largest refugee camps in Jordan, they host about 125,000 Syrian refugees [3]. International Medical Corps (IMC) has played a pivotal role in providing healthcare services in both camps, operating clinics and mobile medical units to address the health needs of camp residents. During the COVID-19 pandemic, IMC implemented several strategies in aspect of infection prevention and control measures that were mandated by the National Defence Law and the United Nations High Commissioner for Refugees (UNHCR). These strategies ensured conducting COVID-19 testing and case management and promoting vaccination campaigns to mitigate the spread of COVID-19 [4,5]. Implemented measures involved temperature screening at entry points of the camp and hospitals, active surveillance and contact tracing, social distancing and gatherings prevention, direct contact isolation,

strict follow up, preparing and upgrading isolation areas, as well as ensuring good practices in providing healthcare services and providing personal protection equipment [6]. Jordan's efforts at the national and local levels, including the activation of the Defence Law and collaboration with health authorities, aimed to combat the spread of COVID-19 [7]. However, the distinctive challenges faced by refugees necessitate focused investigation to ensure that public health interventions are adequately tailored to their needs. It is helpful to evaluate the preparedness and response strategies within refugee camps and contribute valuable insights toward safeguarding these vulnerable communities against the adverse effects of COVID-19 and future health crises [8]. Understanding and investigating the incidence of infections in humanitarian settings is critical to control disease. Exploring incidence in this context could help the global community to grant equitable access to healthcare and resources for the refugees' communities, and governments and authorities to legislate appropriate strategies and policies to control future crisis in those communities. However, there's scarcity of what is known about the incidence of the COVID-19 in camp population in Jordan. The lack of such knowledge restricts mitigating the impact of future pandemic and suggesting appropriate intervention.

A retrospective study was conducted in Greece to estimate the incidence and outbreaks of COVID-19 among resident asylum seekers and refugees in camps across 9 months period. The results indicated an incidence of 1758- 2052 per 100,000 in those communities, which was almost three times higher than the general population of Greece. In Jordanian context, only one study investigated the incidence among Zaatari and Azraq camps and found that the incidence rates were lower in the camps compared to the general population of the country [8]. They suggested that follow-up research on COVID-19, investigating the trend of infection rate over time, the appropriateness of mitigation and public health strategies, and the effect on the healthcare system, is paramount to understand the case in such humanitarian settings. However, this previous study covered a limited period, namely during the first and second waves of alpha and beta variants, prior to the distribution of COVID-19 vaccines in the camps. Therefore, there is still a pressing need for longitudinal studies that capture temporal trends and variations over an extended period, covering different variants of COVID-19 and capturing the effect of vaccination. The current two-year study, covering the four waves of COVID-19 cases, aimed to thoroughly examine the epidemiological characteristics, incidence rates, case detection, isolation, vaccination, risk determinants, and outcomes of COVID-19 among the Syrian refugees residing in Azraq Camp, Jordan. The study seeks to inform targeted public health interventions and policies that can effectively address the needs of refugees during outbreaks or pandemics.

## Methods

### Ethics statement

This retrospective observational study investigated the epidemiological characteristics of COVID-19 among Syrian refugees in Azraq Camp. The study population consisted of all Syrian refugees who resided in Azraq Camp at the time of the study and at least had one positive COVID-19 test between August 1, 2020, and August 31, 2022, recurrent cases were retained, as they may provide a deeper understanding of the incidence across different waves and offer a broader perspective on the progression of the virus over time. Ethical permission to conduct this study was obtained from the Human Research Ethics Committee at Jordan University of Science and Technology (IRB #: 24/162/2023). Consent forms were waived as the study utilized retrospective, de-identified data, and no direct contact with participants was involved.

## Study settings

Jordan now supports a significant population of refugees and asylum seekers, comprising around 760,000 individuals who have been officially registered with the United Nations High Commissioner for Refugees (UNHCR) [9]. Of the total number of refugees in Jordan, around 670,000 are Syrians. The United Nations offers a range of services to refugees inside camps and in host communities. These services include many forms of support, such as protection, food and monetary aid, and primary healthcare, among other essential provisions [9]. The Azraq refugee camp is situated roughly 100km east of Amman in the Zarqa Governorate and commenced operations in April 2014. The camp serves as a residence for an estimated population of more than 40,000 refugees. The camp consists of only four villages numbered as villages 2, 3, 5, and 6. The population of each village ranges between 10,000 and 12,000 residents [4]. Villages 2,3, and 6 were directly interconnected, whereas village 5 was entirely isolated as a strategy of suppressing the virus.

## Data collection

Data were collected through a COVID-19 community case management program, implemented by IMC. This program was designed to alleviate the heavy load the pandemic has placed on healthcare system in the camp by providing support for home-based care of COVID-19 patients. It included initiatives for case investigation, contact tracing, follow-up care, and medical monitoring. The program was carried out by four community health teams supervised by a community health general practitioner. Each team consisted of one community health nurse and ten community health workers.

Data gathering involved telephone interviews or face-to-face meetings during home visits, executed by trained members of these health teams. The collected data, which were then entered into an online database for subsequent analysis, included socio-demographic characteristics of the patients, comorbidities, history of SARS-CoV-2 testing along with results, a list of close contacts and their locations, and detailed vaccination profiles. To ensure the data's accuracy and completeness, community health nurses conducted audits on all the data collected. Moreover, IMC ensured continued follow-up with all participants through calls or home visits to monitor any new symptoms or the progression of any complications.

## Study outcome measures

The primary outcomes included the incidence rate of COVID-19 cases within the camp, analyzing how these rates fluctuated over time to identify any trends. Additionally, the study examined the clinical outcomes of these cases to gauge the severity and recovery rates among the refugee population. The secondary outcome was to identify the risk factors of the severity of COVID-19 infections and hospitalization.

## Statistical analysis

Data were analyzed using STATA statistical software version 16.0 (Statacorp, College Station, Texas, USA).The statistical analyses included descriptive statistics to describe the study population and the epidemiological characteristics of COVID-19 cases, over time and several groups. The categorical variables were presented as frequencies and percentages, while the continuous variables were presented as medians and interquartile ranges (IQRs). Differences in percentages were assessed using Chi-square test. We estimated the monthly incidence rate of COVID-19 cases per 100,000 population of the camp and compared it with Jordan's country incidence rate over the study period. The monthly incidence risk ratio (IRR) of COVID-19

cases inside the camp were plotted and compared with the monthly number of cases in Jordan using Poisson regression. We also estimated and plotted the monthly incidence of cases per 1000 individuals across genders, age groups, villages, and case severity. We plotted the monthly trend of all study characteristics and reported the IRRs and 95% confidence intervals (95% CIs) assuming Poisson distribution.

To identify the risk factors of hospitalization (i.e., including those who were hospitalized or died), we conducted multilevel logistic random-effect regression models clustered by village. Differences in incidence and public heath interventions among the villages of the camp were noted during the analysis. We estimated the adjusted odds ratio (aOR) and 95% CIs after including and controlling for age groups, sex, comorbidities, vaccination status, and calendar month. The association between variables and hospitalization was considered statistically significant when the p-value was less than 0.05.

## Results

### Incidence

A total of 2,468 confirmed COVID-19 cases were recorded during the study period, culminating in a cumulative prevalence rate of 5.6% across the total camp population (43,999 refugees) over the study period.

Fig 1 presents a comparative analysis of the monthly COVID-19 incidence rate per 100,000 persons in Azraq refugee camp alongside national statistics for Jordan. The monthly crude incidence rate was lower in Azraq refugee camp compared to the national level incidence by more than 50% for almost all months of the study (Overall IRR: 0.420, 95% CI: 0.404 to 0.437, p-value<0.001), except in September and October and in May 2022. The monthly incidence rate in the camp reduced significantly by an average of 1.7% (IRR: 0.983; 95% CI:0.978 to 0.988, p-value < 0.001).

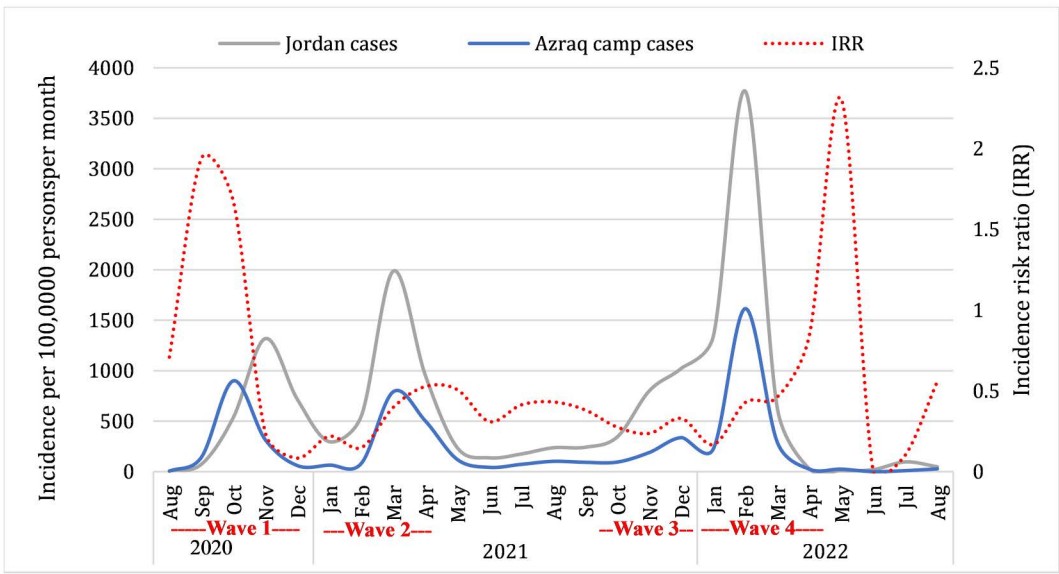

**Fig 1. Comparative temporal trends of monthly COVID-19 incidence rates in Azraq Camp (blue line) and Jordan (gray line) and incidence risk ratios (red dotted line reflects the relative incidence rates between Azraq camp and the national population).**

Table 1 presents the distribution of COVID-19 cases in the Azraq camp with respect to gender, age group, and village of residence within camp. Our data revealed a notable gender difference in COVID-19 cases, with females accounting for 56.4% of cases and showing a higher cumulative incidence rate (6.4 cases per 100 females) compared to males (4.9 cases per 100 males). These differences remained consistent in most months and without a significant change in the monthly incidence (Fig 2A). Older individuals bore the most substantial burden of disease, particularly those aged over 60 years (cumulative incidence: 12.1%) and between those aged 36 and 59 years (cumulative incidence: 10.4%). Those aged 60 years or older had the highest monthly incidence rate during the first two waves but not the following waves, declining by an average of 5.0% (IRR: 0.950; 95% CI: 0.921–0.979; p=0.001). Individuals aged between 18 and 35 years had a higher monthly incidence rate during the third and fourth waves (Fig 2B), increasing by an average of 1.3% (IRR=1.013; 95% CI: 1.002–1.023; p=0.016).

Our spatial analysis showed that village 5 had the highest cumulative incidence rate of 6.9% and monthly incidence rate during waves one, three, and four. The monthly incidence for all villages varied across the waves (Fig 2C), declining in village 2 by 3.3% (IRR:0.967; 95% CI: 0.955–0.979; p<0.001), and increasing in village 6 by 4.4% (IRR: 1.044; 95% CI: 1.030–1.059; p<0.001).

**Table 1. Demographic characteristics of the COVID-19 cases within Azraq refugee camp.**

| | | COVID-19 cases n (%) | Total camp Population* | Cumulative incidence per 100 refugees | p-value | Monthly incidence change IRR (95% CI) | p-value for trend |
|---|---|---|---|---|---|---|---|
| **Overall** | | 2,468 | 43,999 | 5.6 | | 0.983 (0.978–0.988) | <0.001 |
| **Gender** | Females | 1,393 (56.4) | 21,856 | 6.4 | <0.001 | 0.998 (0.990–1.006) | 0.614 |
| | Males | 1,075 (43.6) | 22,143 | 4.9 | | 1.002 (0.992–1.011) | 0.735 |
| **Age group (years)** | 0–4 | 153 (6.2) | 7,962 | 1.9 | <0.001 | 0.979 (0.956–1.003) | 0.090 |
| | 5–11 | 356 (14.4) | 11,476 | 3.1 | | 0.987 (0.971–1.003) | 0.100 |
| | 12–17 | 264 (10.7) | 6,666 | 4.0 | | 0.999 (0.981–1.018) | 0.912 |
| | 18–35 | 860 (34.8) | 9,983 | 8.6 | | 1.013 (1.002–1.023) | 0.016 |
| | 36–59 | 736 (29.8) | 7,091 | 10.4 | | 1.002 (0.991–1.013) | 0.787 |
| | 60+ | 99 (4.0) | 821 | 12.1 | | 0.950 (0.921–0.979) | 0.001 |
| **Village** | 2 | 580 (23.5) | 10,171 | 5.7 | <0.001 | 0.967 (0.955–0.979) | <0.001 |
| | 3 | 599 (24.3) | 11,802 | 5.1 | | 1.005 (0.993–1.018) | 0.412 |
| | 5 | 775 (31.4) | 11,208 | 6.9 | | 0.991 (0.981–1.002) | 0.104 |
| | 6 | 504 (20.4) | 10,818 | 4.7 | | 1.044 (1.030–1.059) | <0.001 |

*All registered refugees.

A. Gender

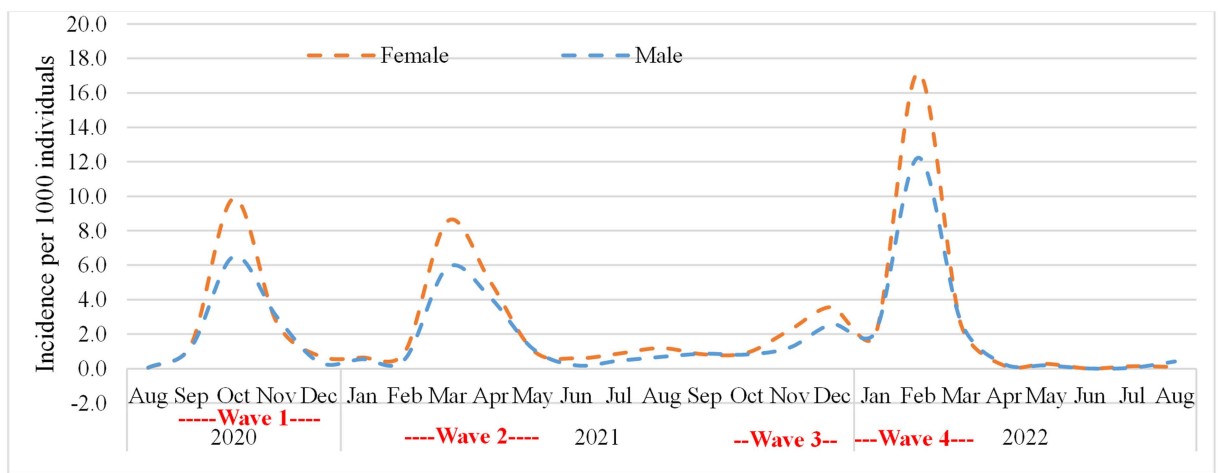

B. Age groups

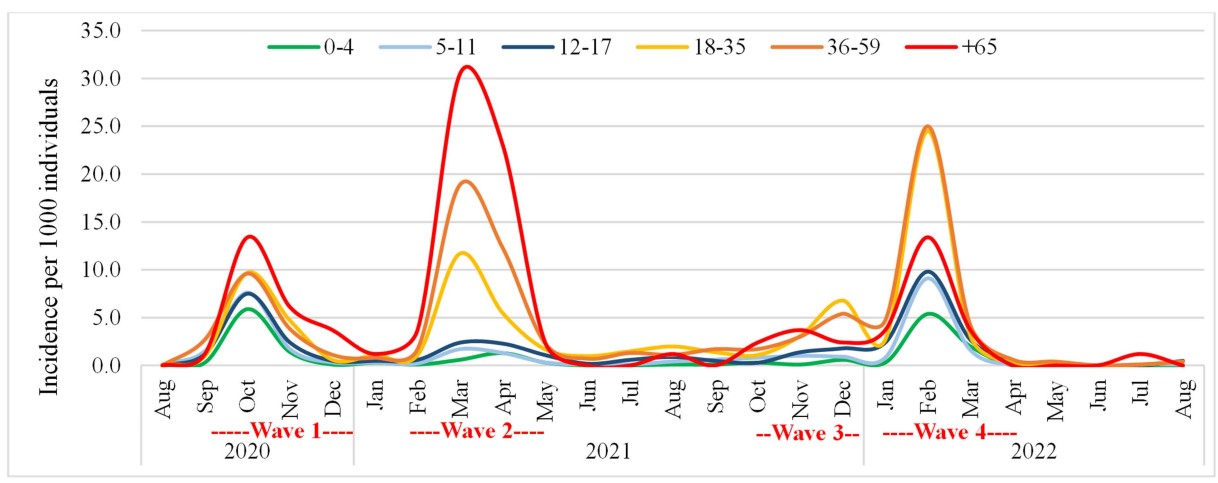

C. Residence

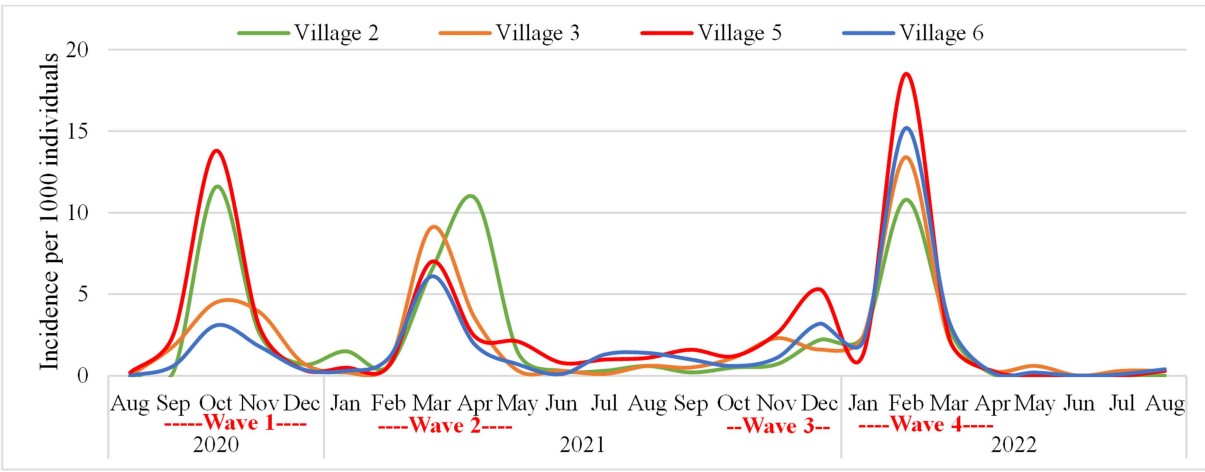

**Fig 2. Trends of incidence per 1000 individuals over time by gender (A), age groups (B), and residence (C).**

## Detection, Isolation, and vaccination

Table 2 outlines the strategies for COVID-19 detection, contact tracing, isolation, and vaccination efforts within the Azraq refugee camp. There were four main ways that cases could have been funneled to COVID-19 testing.The primary mode of detection was individuals seeking healthcare (52.23%), who could have appeared at a hospital or medical office for a visit, either with COVID-19 symptoms or for another reason, and the hospital or physician may have ordered a test. The second mode of action was population surveillance (36.14%), who could have been randomly selected for COVID-19. The third mode of detection was at point of entry (9.08%), as COVID-19 testing was required for all travelers at points of entry to the camp. Finally, only 2.55% of the cases could have been identified as part of the contact tracing process, who were a household or non-household close contact, by a recent interviewed case and instructed to undergo COVID-19 testing. The variation in COVID-19 detection methods over the two-year span shows distinct patterns for each method (Fig 3A). The monthly rate of those who were detected after seeking health care increased over the study period at a rate of 4.1% (95% CI: 3.2%–5.0%; p-value<0.001), while the monthly rate of those cases who were detected by surveillance reduced at a rate of 3.8% (95% CI: 2.9–4.8%; p-value<0.001). Notably, 13.9% of cases had left the camp, and 22.2% had visited a healthcare facility within 14 days prior to symptom onset, while 9.20% had been in contact with a confirmed case.

Regarding response efforts, the median number of contacts (individuals who were in contact with the study confirmed cases) per case was 6 (IQR=4–7), with a similar follow-up rate. Isolation was enforced in various settings, with the majority in home-based isolation (57.3%) or Azraq camp public areas (30.1%). Home-based isolation strategy was the most dominant

**Table 2. COVID-19 Detection and Response Statistics in Azraq Refugee Camp.**

| | n (%) | Trend over time IRR (95% CI) | p-value |
|---|---|---|---|
| **Detection Type** | | | |
| Seeking health care | 1,289 (52.2) | 1.041 (1.032–1.050) | <0.001 |
| Surveillance (including active surveillance) | 892 (36.1) | 0.962 (0.952–0.971) | <0.001 |
| Point of entry | 224 (9.1) | 0.977 (0.958–0.997) | <0.001 |
| Contact-tracing | 63 (2.6) | 0.813 (0.765–0.864) | <0.001 |
| **Was out of the camp within 14 days prior to symptom onset** | 343 (13.9) | 0.974 (0.958–0.990) | 0.001 |
| **Visited health facility within 14 days prior to symptom onset** | 548 (22.2) | 1.034 (1.021–1.048) | <0.001 |
| **Contacted with a confirmed case prior to symptom onset** | 227 (9.2) | 0.901 (0.881–0.922) | <0.001 |
| **Number of contacts (total=13,307), median (IQR)** | 6 (4–7) | 0.986 (0.984–0.989) | <0.001 |
| **Number of contacts followed (traced, total=13,214), median (IQR)** | 6 (4–7) | 0.986 (0.983–0.988) | <0.001 |
| **Number of individuals isolated (total=9640), median (IQR)** | 4 (1–6) | 0.988 (0.986–0.991) | <0.001 |
| **Isolation location** | | | |
| Home-Based isolation | 1,414 (57.3) | 1.007 (0.999–1.015) | 0.087 |
| Azraq Camp (Public area) | 743 (30.1) | 0.981 (0.971–0.992) | 0.001 |
| Hospital | 251 (10.2) | 1.007 (0.988–1.027) | 0.448 |
| Out of the camp | 60 (2.4) | 1.016 (0.977–1.057) | 0.428 |
| **Vaccinated before infection (self-reported)** | 780 (31.6) | 1.218 (1.200–1.241) | <0.001 |
| **Type of vaccine among vaccinated** | | | |
| Pfizer | 389 (49.8) | 1.28 (1.240–1.320) | <0.001 |
| Sinopharm | 357 (45.8) | 1.178 (1.150–1.207) | <0.001 |
| AstraZeneca | 34 (4.4) | 1.161 (1.087–1.240) | <0.001 |

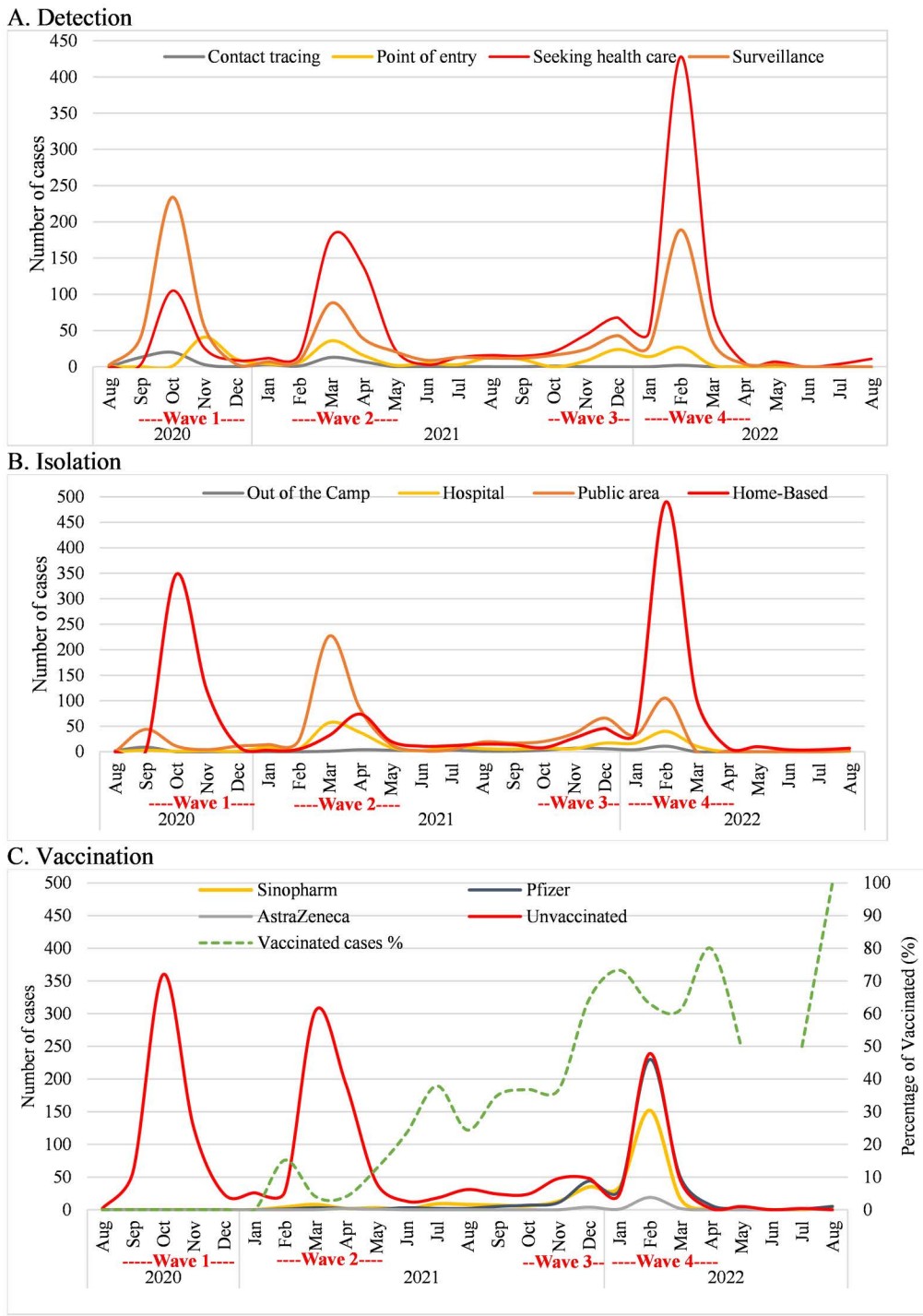

**Fig 3. The number of cases over time by (A) detection methods, (B) isolation approaches, and (C) vaccination status at time of infection.**

during the study period except for the second wave when the public area isolation was the dominant approach (Fig 3B). Isolation in a public space strategy declined (IRR: 0.981; 95% CI: 0.971–0.992; p-value=0.001) over the study period while home-based isolation tended to show a slight increase (IRR: 1.007; 95% CI: 0.999–1.015; p-value=0.087).

Vaccination prior to infection was self-reported by 31.6% of the cases, with Pfizer being the most common vaccine administered (49.8%), followed by Sinopharm (45.8%), and AstraZeneca (4.4%). The median duration since the last vaccine dose was 140 (IQR=54–229) days. The rate of vaccinated cases increased by 21.8% (IRR:1.218; 95% CI:1.200–1.241; p-value<0.001) per month as depicted in Fig 3C.

## Case severity and risk factors

Of total cases, 60.6% (n=1,496) were asymptomatic. During the follow-up period of 14 days, the severity of cases was categorized as follows: 56.0% (n=1,383) were asymptomatic, 32.8% (n=819) displayed symptoms without requiring hospitalization, 10.4% (n=257) necessitated hospital admission, and only 0.8% (n=19 cases) resulted in fatalities.

Fig 4 charts the evolution of case severity within the camp during the study period. Asymptomatic cases dominated mostly but the monthly rate decreased over time by an average of 4.5% (IRR: 0.955; 95% CI: 0.947–0.963), while monthly symptomatic cases gradually increased by an average of 8.2% (95% CI: 1.069–1.095; p-value<0.001). Hospitalization rates varied and increased by an average of 2.0% (IRR: 1.020; 95% CI: 1.001–1.040; p-value=0.037), with notable surges aligning with the second and third case spikes, suggesting correlation with infection

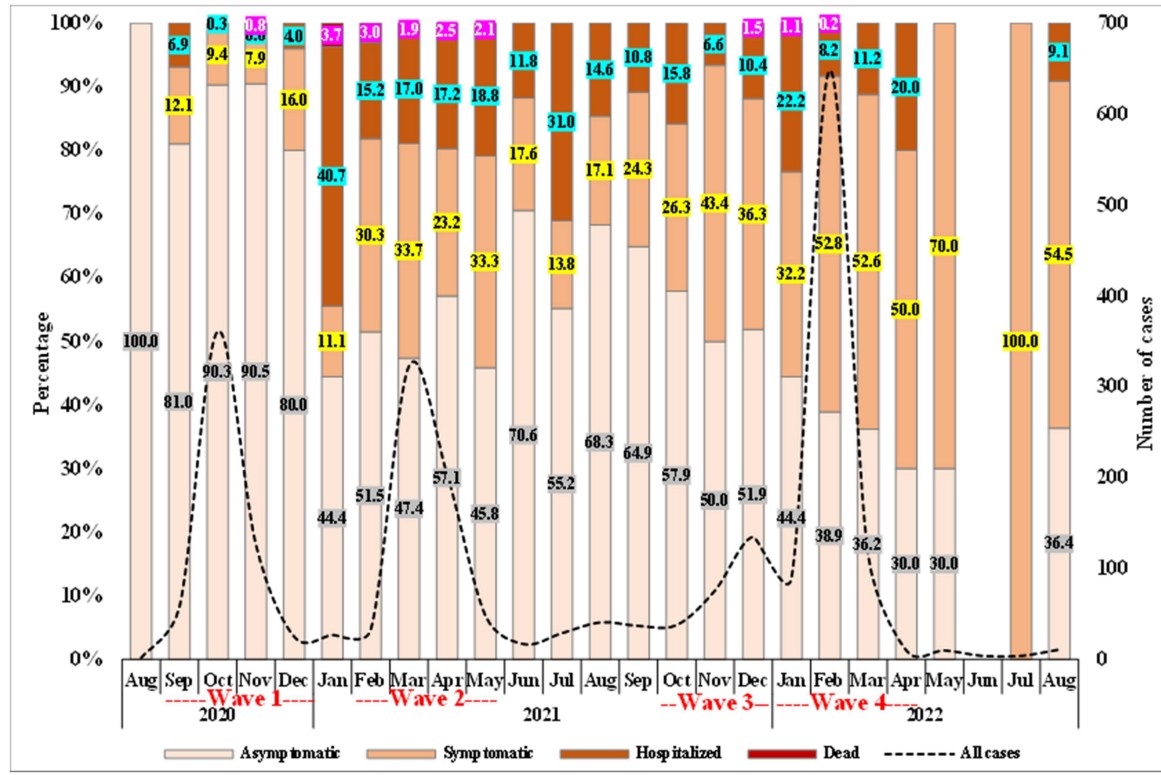

**Fig 4. Distribution of COVID-19 cases during the study period by case severity.**

rates. The monthly death rate remained a steadily small fraction of total cases (IRR: 0.953: 95% CI: 0.888–1.022; p-value=0.175).

Table 3 presents a descriptive analysis of the chi-square test for the COVID-19 case severity in relation to demographics and comorbidities. Case severity significantly increased with older age groups but did not differ between male and female sexes. There was a significant difference between case severity and comorbidities; confirmed COVID-19 cases with any comorbidities had nearly double the hospitalization (19.8% vs 8.3%) and more than four times the death percentage compared to those without any comorbidities (2.2% vs 0.5%, respectively).

**Table 3. COVID-19 case severity by demographic characteristics, comorbidities, pregnancy, and vaccination status in Azraq camp, Jordan.**

| | Asymptomatic | Symptomatic outpatients | Hospitalized | Dead | Total | p-value [a] |
|---|---|---|---|---|---|---|
| | n (%) | n (%) | n (%) | n (%) | n | |
| **Age group** | | | | | | <0.001 |
| 0-4 | 96 (62.7) | 44 (28.8) | 12 (7.8) | 1 (0.7) | 153 | |
| 5–11 | 232 (65.2) | 106 (29.8) | 18 (5.1) | 0 (0) | 356 | |
| 12–17 | 142 (53.8) | 106 (40.2) | 16 (6.1) | 0 (0) | 264 | |
| 18–35 | 486 (56.5) | 277 (32.2) | 96 (11.2) | 1 (0.1) | 860 | |
| 36–59 | 398 (54.1) | 249 (33.8) | 82 (11.1) | 7 (1.0) | 736 | |
| 60+ | 29 (29.3) | 27 (27.3) | 33 (33.3) | 10 (10.1) | 99 | |
| **Gender** | | | | | | 0.895 |
| Female | 781 (56.1) | 452 (32.4) | 150 (10.8) | 10 (0.7) | 1393 | |
| Male | 602 (56.0) | 357 (33.2) | 107 (10.0) | 9 (0.8) | 1075 | |
| **Village** | | | | | | <0.001 |
| Village 2 | 324 (55.9) | 168 (29.0) | 82 (14.1) | 6 (1.0) | 580 | |
| Village 3 | 272 (45.4) | 255 (42.6) | 65 (10.9) | 7 (1.2) | 599 | |
| Village 5 | 549 (70.8) | 178 (23.0) | 47 (6.1) | 1 (0.1) | 775 | |
| Village 6 | 230 (45.6) | 206 (40.9) | 63 (12.5) | 5 (1.0) | 504 | |
| **Mode of detection** | | | | | | <0.001 |
| Seeking health care | 354 (27.5) | 711 (55.2) | 212 (16.4) | 12 (0.9) | 1289 | |
| Surveillance | 804 (90.1) | 62 (7.0) | 21 (2.4) | 5 (0.6) | 892 | |
| Point of entry | 180 (80.4) | 27 (12.1) | 15 (6.7) | 2 (0.9) | 224 | |
| Contact tracing | 45 (71.4) | 9 (14.3) | 9 (14.3) | 0 (0) | 63 | |
| **Comorbidities** | | | | | | |
| Without comorbidities | 1188 (59.2) | 645 (32.1) | 166 (8.3) | 9 (0.5) | 2008 | <0.001 |
| With comorbidities | 195 (42.4) | 164 (35.7) | 91 (19.8) | 10 (2.2) | 460 | <0.001 |
| CVD | 70 (41.9) | 60 (35.9) | 33 (19.8) | 4 (2.4) | 167 | <0.001 |
| Diabetes | 43 (44.3) | 31 (32.0) | 21 (21.7) | 2 (2.1) | 97 | 0.001 |
| **Comorbidities among women** | | | | | | <0.001 |
| No risk factors | 656 (60.5) | 329 (30.4) | 92 (8.5) | 7 (0.7) | 1084 | |
| Any risk factors | 125 (40.5) | 123 (39.8) | 58 (18.8) | 3 (1.0) | 309 | |
| **Pregnancy among women** | | | | | | 0.003 |
| Pregnant | 53 (41.7) | 54 (42.5) | 20 (15.8) | 0 (0) | 127 | |
| Non-pregnant | 728 (57.5) | 398 (31.4) | 130 (10.3) | 10 (0.8) | 1266 | |
| **Vaccinated** | | | | | | <0.001 |
| No | 1057 (62.6) | 445 (26.4) | 169 (10.0) | 17 (1.0) | 1688 | |
| Yes | 326 (41.8) | 364 (46.7) | 88 (11.3) | 2 (0.3) | 780 | |

[a]Chi-square test.

Among women with COVID-19, there was a statistically significant difference in case severity by presence of comorbidities and by pregnancy. Notably, those with any comorbidities had higher percentage that were hospitalized (18.8% vs 8.5%), as did pregnant women compared to women who were not pregnant (15.8% vs 10.3%).

Case severity differed significantly with the geographical distribution (villages) and modes of detection within the camp. Most cases from village 5 were asymptomatic but they had lower percentages of hospitalized and deceased cases compared to those from other villages. Those identified through seeking health care had the highest percentages of symptomatic cases and hospitalizations compared to other modes. Notably, vaccinated cases presented more as symptomatic outpatient cases than unvaccinated cases (46.7% vs 26.4%, respectively), but they had a lower percentage of cases that died (0.2% vs 1.0%, respectively).

## Risk factors for hospitalization

The results of the multivariable model revealed significant differences in risk of hospitalization among various age groups (Table 4). Compared to young adults (aged 18-35), younger age groups, particularly those aged 5–11 and 12–17, had a decreased odds of hospitalization with aOR of 0.35 (95% CI: 0.20-0.61, p-value<0.001) and 0.44 (95% CI: 0.25–0.77, p-value =0.004), respectively. On the contrary, elderly (aged ≥60) had more than a 4-fold increase in the odds of hospitalization (aOR: 4.16; 95% CI: 2.54–6.82; p-value <0.001) compared to young adult cases. Furthermore, cases with comorbidities had almost a two-fold increase in the odds of hospitalization (aOR: 1.95; 95% CI: 1.38–2.74, p-value <0.001). On the contrary, male sex was not a significant predictor of hospitalization risk (aOR: 1.02, 95% CI: 0.78–1.32, p-value =0.963), indicating that the vulnerability to hospitalization is comparable across sexes.

Compared to unvaccinated cases, being vaccinated prior to infection was not associated with hospitalization in the univariable analysis. However, in the multivariable analysis, being vaccinated was associated with a reduction in the odds of hospitalization by 45% (aOR: 0.55; 95% CI: 0.38–0.01; p-value=0.003). The calendar month was associated with an increase in the

**Table 4. Risk factors of hospitalization among COVID-19 cases in Azraq camp.**

|  | Univariable logistic regression | | | Multivariable logistic regression | | |
|---|---|---|---|---|---|---|
|  | OR | 95% CI | p-value | aOR | 95% CI | p-value |
| **Age groups** |  |  |  |  |  |  |
| 0–4 | 0.74 | 0.40–1.35 | 0.326 | 0.62 | 0.32–1.17 | 0.138 |
| 5–11 | 0.43 | 0.25–0.72 | 0.001 | 0.35 | 0.20–0.61 | <0.001 |
| 12–17 | 0.48 | 0.28–0.83 | 0.009 | 0.44 | 0.25–0.77 | 0.004 |
| 18–35 (Reference) | 1 | -- | -- | 1 | -- | -- |
| 36–59 | 1.02 | 0.75–1.39 | 0.886 | 0.93 | 0.67–1.28 | 0.656 |
| 60+ | 5.50 | 3.49–8.67 | <0.001 | 4.16 | 2.54–6.82 | <0.001 |
| **Male sex** | 0.96 | 0.74–1.23 | 0.730 | 1.03 | 0.79–1.35 | 0.805 |
| **With comorbidities** | 2.85 | 2.11–3.85 | <0.001 | 1.95 | 1.38–2.74 | <0.001 |
| **With CVD** | 2.43 | 1.64–3.60 | <0.001 | -- | -- | -- |
| **With DM** | 2.50 | 1.53–4.89 | <0.001 | -- | -- | -- |
| **Women with comorbidities** | 2.14 | 1.43–3.22 | <0.001 | -- | -- | -- |
| **Pregnant among women** | 1.40 | 0.84–2.35 | 0.196 | -- | -- | -- |
| **Vaccinated cases** | 1.06 | 0.81–1.39 | 0.684 | 0.55 | 0.38–0.81 | 0.003 |
| **Calendar month** | 1.02 | 0.99–1.04 | 0.056 | 1.06 | 1.03–1.09 | <0.001 |

CVD: cardiovascular diseases, DM: diabetes mellitus.

odds of hospitalization (aOR: 1.06, 95% CI: 1.03–1.09, p-value<0.001), indicating the risk of hospitalization increased by 6% for every calendar month. The significant likelihood ratio test (p<0.001) confirms that incorporating village-level random effects into the model provides a better fit to the data than a model ignoring these cluster effects.

## Discussion

This study investigated the epidemiological characteristics of the disease frequency and case severity of COVID-19 in Azraq refugee camp. The incidence rate in the camp was lower than the national level by more than 50%. The incidence rate was higher among females, the elderly, and refugees residing in village five. Most detections occurred through healthcare-seeking, with home-based isolation as the main response strategy Vaccination rates grew to cover 31.6% of cases, predominantly with the Pfizer vaccine. Outpatient symptomatic cases made up about half of cases, while one in every 10 cases were hospitalized. Age, location, detection methods, underlying conditions, pregnancy, and vaccination status independently associated with disease severity.

A previous study recorded 901 cases in Azraq and 1,715 in Zaatari up to April 2021, with incidence rates lower in camps than neighboring governorates and Jordan overall [8]. This lower incidence persisted in the Azraq camp beyond the first six months of the pandemic, with monthly rates consistently lower than the national average throughout the two-year study period [8]. This lower incidence persisted in the Azraq camp beyond the first six months of the pandemic, with monthly rates consistently lower than the national average throughout the two-year study period [8]. In contrast to our results, the analysis of COVID-19 incidence among refugees and asylum seekers in densely populated receiving centers in Greece revealed higher rates compared to the general population [8,10]. In contrast to our results, the analysis of COVID-19 incidence among refugees and asylum seekers in densely populated receiving centers in Greece revealed higher rates compared to the general population [10]. This discrepancy can be attributed to the notably poorer living conditions within these facilities (characterized by inadequate sanitation, limited access to healthcare services, and a shortage of essential resources) or to the higher population density, frequent movements, severe overcrowding, and difficulties in implementing social distancing and hygiene measures [10,11]. Conversely, Uganda's refugee settlements exhibited lower incidence rates than the national average, although limited testing capacity within these settlements may have contributed to variations against the national level and across different settlements [12]. The testing rates in the Azraq camp exceeded the national average by almost 70%, indicating proactive surveillance efforts within the camp [8,12]. The testing rates in the Azraq camp exceeded the national average by almost 70%, indicating proactive surveillance efforts within the camp [8].

The initial months of the first wave in Azraq camp had higher incidence rates in the camp compared to the national average, albeit with a shorter duration of the wave within the camp. This could be due to increased testing and detection efforts during the early phase of the pandemic. Early implementation of nonpharmaceutical intervention measures, such as social distancing, curfew, and mandatory mask-wearing, and applying the preparedness and response strategies likely contributed to the delayed introduction and reduced transmission of COVID-19 among the refugee population [13]. Therefore, our findings support the effectiveness of these interventions and emphasize the importance of sustained public health measures and robust disease surveillance systems to mitigate the pandemic's impact on vulnerable populations in dense living conditions.

The incidence rate estimates within the four villages of Azraq camp differed and ranked inconsistently during the study period. For example, village 6 had the lowest incidence in the first and second waves then ranked the second in the following two waves. This might be

related to the population immunity, or variation in testing or surveillance capacities within the villages. In a background analysis of our data, the percentage of detection modes differed significantly within the villages. Village 5 had reported the largest proportion of cases detected through surveillance (>60%), which justifies the isolation of the village from the other neighboring villages during the pandemic outbreak, indicating that the authorities were closely monitoring the epidemiological situation in the camp and promptly implementing immediate, appropriate actions. However, village 2 had the largest proportion of cases detected when they sought health care (>40%). This highlights the difference in the distribution of surveillance teams and health care services within camp villages. The variation in detection methods and resource allocation, such as human resources, transport, tests, and supplies, could explain these observed differences.

The higher COVID-19 incidence rate among females in the Azraq camp is contrary to the global trends favoring males [12,14–17]. Given that men in refugee camps in Jordan are more likely to work than women, one explanation of this discrepancy might stem from economic concerns that lead men to underreport infections as they are mandated to be isolated for 14 days, losing daily income [18]. In addition, men may be less likely to be tested during the surveillance and contact-tracing as they are usually not available at home during working hours [12,14–17]. Given that men in refugee camps in Jordan are more likely to work than women, one explanation of this discrepancy might stem from economic concerns that lead men to underreport infections as they are mandated to be isolated for 14 days, losing daily income [18]. In addition, men may be less likely to be tested during the surveillance and contact-tracing as they are usually not available at home during working hours. Women, on the other hand, may seek healthcare services more often due to their roles in the household and higher healthcare awareness, leading to higher detection rates, thus, another explanation might be the higher rate of women seeking healthcare as most of the cases in the camp were detected after seeking health care. This underscores the importance of context-sensitive public health strategies considering behavioral factors, such as policies protecting sick leave and support for families losing income during isolation. We suggest that public health activities consistently compare the testing rates between males and females and consider surveillance and contact tracing efforts in the workplaces.

Regarding case severity, most of the infected cases were asymptomatic even after a 14-day follow-up. However, the monthly rate of asymptomatic cases declined by 4.5%, while the symptomatic and hospitalized cases increased by 8.2% and 2.0%, respectively. Among infected patients in the camp, 10.4% necessitated hospital admission. A notable increase in the hospitalization rate with the overall infection rate suggests a relationship between the infection and hospitalization rates. Those who sought healthcare were more likely to have severe symptoms compared to other modes of detection, which may have contributed to an overrepresentation of severe cases among those tested positive for COVID-19. Our multivariable regression model revealed several factors associated with higher odds of hospitalization including older age, comorbidities, and the calendar month. On the other hand, patients in younger age, and those vaccinated prior to infection were associated with a reduction in hospitalization rate. Elderly patients were more likely to be hospitalized compared to younger cases. This is in line with other global, regional, and local studies, including Altare et al [8] who found a higher risk of hospitalization among the elderly residing in Zaatari and Azraq camps [19–22]. Furthermore, cases with comorbidities had almost a 2-fold increase in the odds of hospitalization. This pattern mirrors local and global findings, highlighting a significant correlation between pre-existing health conditions and increased risk of severe COVID-19 manifestations, hospitalizations, and fatalities [15,17,23,24].

Notably, being vaccinated prior to infection was independently associated with a reduction in the adjusted odds of hospitalization by 45%, highlighting the possible importance of COVID-19 vaccination as a public health strategy in such humanitarian settings. This is lower than what was reported by a previous systematic review and meta-analysis where the hospitalization rate was reduced by about 73% (95% CI 95–82%) among those being vaccinated [25]. The lower odds of hospitalization among vaccinated camp residents should be attributed to the study design, such as the study period, which started a year before the vaccination initiative commenced. However, little is known about the effectiveness and safety of different COVID-19 vaccine types among refugees, thus, shedding light on this gap is critical for future research to further understand the role of vaccination as a public health strategy in the refugee camps and to guide future implementation of those strategies.

The burden of COVID-19 and its variants of concern varied significantly across the four waves in the Azraq camp as it varied locally and globally. The first wave, driven by the Alpha variant, peaked in October 2020, with a lower rate of hospitalization and less reliance on healthcare-seeking for detection. However, this wave reached its peak locally after a month compared to the camp [26]. The second wave, associated with the Beta variant, was more aggressive, peaking in mid-March 2021 with increased hospitalizations and mortality rate. The pattern of this wave within the camp was similar to the pattern observed in rest of the country [16]. Pre-Delta variant (Jun-Oct 2021) demonstrated less infection rate with a decrease in hospitalization and no death recorded in this period. Like the pattern in Jordan, the third wave of viral spread increased rapidly in mid-October 2021 and reached the peak in mid-December 2021, with the number of daily reported cases starting to decline progressively afterwards. Despite its higher incidence, it was less severe than the Beta wave. This could indicate the possible effectiveness of COVID-19 vaccination campaigns in the camp. The fourth wave, associated with the Omicron variant, started in January 2022, and was described to be embedded within the previous wave. Despite its high transmissibility, its severity varied from January 2022 onward, with some reduction in hospitalization and mortality rates compared to previous waves. This reduction could be attributed to increased immunity from vaccinations and prior infections, improved treatment protocols, and the implementation of nonpharmaceutical interventions. These characteristics of Omicron variant align with findings from other local and global studies, where its higher infection rates were associated with varied symptoms and lower hospitalization rates [27].

## Limitations

Despite the valuable insights provided by this study, it is essential to acknowledge several limitations. Firstly, as this research relies on secondary data analysis for a single camp, its findings are subject to the quality and comprehensiveness of the data collected. Moreover, the use of self-reported data may introduce biases and inaccuracies, potentially influencing the validity of the results. For example, there might be underreporting of contacts and infections due to fear of isolation and stigma or overreporting of vaccination status due to social desirability bias. Additionally, discrepancies between the total population of the camp and the actual number of residing refugees may impact the generalizability of the findings, and the interpretation of the epidemiological characteristics observed. We also did not account for periods of immunity after being infected as the denominator remained consistent over the study period, however, individuals with multiple infections were observed over the two-year study period, who were included in the analysis. Future research should aim to address these gaps by fostering collaboration with relevant stakeholders, including camp authorities (i.e., UNHCR) and international organizations, to gain access to comprehensive data on population dynamics. Further, this study did not explore the psychological impacts of the pandemic on refugees, the long-term implications of

COVID-19 on their health and well-being, or more qualitative insights from residents which could provide a deeper understanding of their experiences and challenges. Further research is needed to explore these gaps using an appropriate research design. Finally, for factors associated with case severity, other unmeasured factors may influence the observed associations such as socioeconomic status, warranting caution in drawing definitive conclusions.

## Conclusion

This study offers critical insights into the epidemiological dynamics of COVID-19 among Syrian refugees residing in Jordan's Azraq camp. Our findings reveal that the crude incidence rate was lower in Azraq camp in comparison to the national level incidence by more than 50% for almost all months of the study with a significant 1.7% monthly decrease in the incidence rate within the camp. Potential benefits of proactive measures such as surveillance, contact tracing, isolation, and vaccination campaigns could be found by this study. Certain factors such as age, residency location, underlying health conditions, and vaccination status significantly influence the severity of COVID-19 cases among camp residents. These findings highlight the imperative need for comprehensive healthcare strategies tailored to the constraints and needs of camp populations. There may be some room for optimal response to future epidemics, such as initiating robust health surveillance, vaccination programs, and allocating resources for chronic disease management which may better mitigate COVID-19's impact and bolster overall public health resilience in refugee camp settings.

## Author contributions

**Conceptualization:** Ahmad Waleed Zghool, Ahmad Alrawashdeh, Khalid Kheirallah.

**Data curation:** Ahmad Waleed Zghool.

**Formal analysis:** Ahmad Alrawashdeh.

**Methodology:** Ahmad Waleed Zghool, Ahmad Alrawashdeh.

**Validation:** Sara A. Nasser, Natalya Kostandova, Shiromi M. Perera, Jomana W. Alsulaiman, Adi H. Khassawneh, Abdel-Hameed W Al-Mistarehi, Amer Abu-Shanab, Khalid Kheirallah.

**Writing – original draft:** Ahmad Waleed Zghool, Ahmad Alrawashdeh, Zaid I Alkhatib.

**Writing – review & editing:** Ahmad Alrawashdeh, Sara A. Nasser, Natalya Kostandova, Shiromi M. Perera, Jomana W. Alsulaiman, Adi H. Khassawneh, Abdel-Hameed W Al-Mistarehi, Amer Abu-Shanab, Khalid Kheirallah.

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
