## [Decision Letter · Decision Letter 0]

13 Nov 2024

PNTD-D-24-01204Temporal trends in the incidence and case severity of COVID-19 cases among the Syrian refugees in Azraq camp in Jordan: a retrospective observational studyPLOS Neglected Tropical Diseases Dear Dr. Alrawashdeh, Thank you for submitting your manuscript to PLOS Neglected Tropical Diseases. After careful consideration, we feel that it has merit but does not fully meet PLOS Neglected Tropical Diseases's publication criteria as it currently stands. Therefore, we invite you to submit a revised version of the manuscript that addresses the points raised during the review process. Please submit your revised manuscript within 60 days Jan 12 2025 11:59PM. If you will need more time than this to complete your revisions, please reply to this message or contact the journal office at plosntds@plos.org. Please include the following items when submitting your revised manuscript:* A rebuttal letter that responds to each point raised by the editor and reviewer(s). You should upload this letter as a separate file labeled 'Response to Reviewers '. This file does not need to include responses to any formatting updates and technical items listed in the 'Journal Requirements' section below.* A marked-up copy of your manuscript that highlights changes made to the original version. You should upload this as a separate file labeled 'Revised Manuscript with Track Changes '.* An unmarked version of your revised paper without tracked changes. You should upload this as a separate file labeled 'Manuscript '. If you would like to make changes to your financial disclosure, competing interests statement, or data availability statement, please make these updates within the submission form at the time of resubmission. Guidelines for resubmitting your figure files are available below the reviewer comments at the end of this letter. We look forward to receiving your revised manuscript. Kind regards, Andrei R. Akhmetzhanov, Ph.D.Academic EditorPLOS Neglected Tropical Diseases Mabel CarabaliSection EditorPLOS Neglected Tropical Diseases

Shaden Kamhawi

co-Editor-in-Chief

Paul Brindley

co-Editor-in-Chief

 **Journal Requirements:** **Additional Editor Comments (if provided):****Reviewers' Comments:** Reviewer's Responses to Questions

**Key Review Criteria Required for Acceptance?**

**Methods**

-Are the objectives of the study clearly articulated with a clear testable hypothesis stated?

-Is the study design appropriate to address the stated objectives?

-Is the population clearly described and appropriate for the hypothesis being tested?

-Is the sample size sufficient to ensure adequate power to address the hypothesis being tested?

-Were correct statistical analysis used to support conclusions?

-Are there concerns about ethical or regulatory requirements being met?

Reviewer #1: The article aimed to capture epidemiological characteristics, evaluation of policies enforced and vaccine, risk and outcome of COVID-19 among the Syrian refugees in Azraq camp, Jordan. Collected data include socio-demographic characteristics, comorbidities, COVID-19 test history and results, contact history and vaccination.

Some conditions may need to be considered, such as recurrent cases, population dynamics (in addition to underregistration, other things like inflow/outflow of the population that were managed by external authorities or effect of birth/death by other causes) and underreporting. It sounded a bit strange without mentioning these points.

Especially, recurrent cases are important as they would be correlated to the incidence, and I would recommend to consider the correlation of recurrent cases in all analysis: for example, risk factor analysis may be influenced by the correlation of recurrent cases as it means "double-counted". All results may be changed due to this point. If the author already considered this point, analyses method might be considered and selected correspondingly as well as well-mentioned in the text.

Some comments in the analysis part.

First, comparison test in Table 3 is blurry. Which test were made in comparing which proportion between subgroups?

The unit of incidence varies across analysis. I would say all analysis can use a unit of 1000 subpopulation to describe incidence. Also, how did you estimate and hypothesize the effect of each month. Why and what did you assume the linear trend of change across months to do monthly change analysis? Please mention in the main text. Furthermore, which test or regression did you use for risk facrtor analysis for calculating adjusted odds ratio? Maybe logistic regression?

Moreover, how did you determine final model to include potential confounders? how does the effect of each confounders?

Reviewer #2: Methods are clearly articulated, however, below needs to be addressed -

134: This is a retrospective study. But data were collected “prospectively”. This is confusing. Please clarify.

141: What data collection tool or instrument was used. It should be annexed.

Reviewer #3: (No Response)

**Results**

-Does the analysis presented match the analysis plan?

-Are the results clearly and completely presented?

-Are the figures (Tables, Images) of sufficient quality for clarity?

Reviewer #1: I would assume that the monthly incidence IRR change was calculated using Poission regression, with the calendar month taken into the analysis as continuous variables. If so, why would you assume linearity trend of risk of COVID-19 per each month? As you present some results, could you please provide some interpretation, such as what the increasing/decreasing % mean? Do you think it is really the case, as it is obvious that the cases are recorded like waves?

And it might be better to explain IRR more: some IRR compares between cases in Azraq camp, while others between one month and the next. Maybe its better to determine in the other way as it might be confusing using the term "IRR" in a different manner.

Adjustment of confounders (age, sex, etc.) were not performed in poisson regression? Also, it would be better to have mapping and show relationship between villages (such as the interaction between villages, providing backgrounds of linkages).

Moreover, interpretation of IRR by home-based isolation strategy (Line 242) requires carefulness: it only indicates some evidence, not strong, so it might be dangerous to say "increased" IRR in this section.

Regarding to risk factor analysis: I think the results between behavior and severity is biased: one possible explanation is that they seek for healthcare as they are more severe. So it provides more value if you could mention such bias and how to control them in the main text. Also in the main text, I think you could conduct chi-squared test to compare the positive rates between subcategories to strengthen the findings.

Finally, please fix minor errors in main text: I think Sep and Oct 2020 not 2022 (Line 185) and aOR for hospitalization among vaccinated group compared to non-vaccinated in multivariable analysis (Line 305; I think table 4 is correct, so text should be modified like 95%CI: 0.38-0.81)

Reviewer #2: Nicely written and appropriately presented with sufficient quality.

Reviewer #3: (No Response)

**Conclusions**

-Are the conclusions supported by the data presented?

-Are the limitations of analysis clearly described?

-Do the authors discuss how these data can be helpful to advance our understanding of the topic under study?

-Is public health relevance addressed?

Reviewer #1: The author expands the results of variables of their interest using other study results in discussion part, which sounds relevant to public health interest as the perception of epidemiological characteristics of COVID-19 and the effective strategy against it.

Limitations are well mentioned, including data collection bias, self-reporting bias and generalizability, immunity and hidden factors.

Conclusions highlights the importance of comprehensive healthcare strategy and characteristics of cases among the camp based on the analysis and findings.

However, some interpretation from the examined results were not covered: for example, how do you understand the trend of IRR with some conditions? Please provide the readers with a clear explanation of how this finding does indicate.

Reviewer #2: (No Response)

Reviewer #3: (No Response)

**Editorial and Data Presentation Modifications?**

Reviewer #1: Some comments and points table- and figure-specific that I would recommend to modify:

*Add total number in each subcategories (Table 3).

*in Table 3 p-value: clarify which data to compare (maybe in the text)

*Add total number of cases per month in each figure (or separated table, etc.) such as Figure 4.

*Figure 2A: I think there is no point to exhibit no. of cases and incidence altogether unless there are difference in population between male and female.

*Figure 3: Consider the graph into cumulative number.

*Figure 3C: vaccination figure is missing.

Reviewer #2: (No Response)

Reviewer #3: (No Response)

**Summary and General Comments**

Reviewer #1: As a result, I would recommend for major revision, as some of the premises in this analysis were missing, and the findings are dispersed, the conclusion is not clear, condition would not be well controlled. Methodologies in the analysis could be revised and implemented better.

Please consider the recurrent cases and re-do the analysis, considering the correlation. Also, if the authors would like to mention about monthly IRR change, describe more about linearity and their assumption, findings and its discussions (practical indication, etc.) so that the readers could understand better.

Reviewer #2: Introduction section should address -

84 – 96: Generally, lacks description of evidences from existing literature, specially in other similar settings.

96: How evaluating preparedness and response strategy is relevant this particular epidemiological study?

98 – 106: Rationale of the research is not clear. Why it is essential to investigate the epidemiology of COVID-19 in this particular setting? How it will help global community?

Discussion section:

Please include some sorts of comparison with similar other settings with possible explanations of difference.

Generally, the study provides useful insight.

Reviewer #3: The structure of the article is logical, with a clear progression from introduction to methods, findings, and conclusions. The tone is professional and objective.

While the article is comprehensive, it could benefit from more detailed discussions on the specific healthcare interventions implemented and their effectiveness in the camp.

To enhance the article, the author could include a more in-depth analysis of the specific public health strategies that were successfully implemented during the pandemic in the camp. Additionally, more qualitative insights from residents could provide a deeper understanding of their experiences and challenges.

Strengths include a robust sample size, comprehensive data collection, and a clear methodology. However, a potential weakness may be the reliance on data from one camp, which could limit the generalizability of the findings to other refugee settings.

The article does not deeply explore the psychological impacts of the pandemic on refugees or the long-term implications of COVID-19 on their health and wellbeing. Additionally, it could have addressed the challenges of implementing public health measures in such settings more comprehensively.

PLOS authors have the option to publish the peer review history of their article (what does this mean? ). If published, this will include your full peer review and any attached files.

**Do you want your identity to be public for this peer review?** For information about this choice, including consent withdrawal, please see our Privacy Policy .

Reviewer #1: No

Reviewer #2: **Yes: ** Charls Erik Halder

Reviewer #3: No

---

## [Decision Letter · Decision Letter 1]

28 Jan 2025

Dear Alrawashdeh,

We are pleased to inform you that your manuscript 'Temporal trends in the incidence and case severity of COVID-19 cases among the Syrian refugees in Azraq camp in Jordan: a retrospective observational study' has been provisionally accepted for publication in PLOS Neglected Tropical Diseases.

Best regards,

Andrei R. Akhmetzhanov, Ph.D.

Academic Editor

Mabel Carabali

Section Editor

Shaden Kamhawi

co-Editor-in-Chief

Paul Brindley

co-Editor-in-Chief

The authors responded well to the reviewers, who had no further objections.

Reviewer's Responses to Questions

**Key Review Criteria Required for Acceptance?**

**Methods**

-Are the objectives of the study clearly articulated with a clear testable hypothesis stated?

-Is the study design appropriate to address the stated objectives?

-Is the population clearly described and appropriate for the hypothesis being tested?

-Is the sample size sufficient to ensure adequate power to address the hypothesis being tested?

-Were correct statistical analysis used to support conclusions?

-Are there concerns about ethical or regulatory requirements being met?

Reviewer #1: Response to Response 8: It's okay to use full model, but I was curious if you accounted for potentially increasing standard error (which I guess we have to check) caused by the increased number of variables taking into account.

Reviewer #2: No further comment. Thanks to the author for clarification.

**Results**

-Does the analysis presented match the analysis plan?

-Are the results clearly and completely presented?

-Are the figures (Tables, Images) of sufficient quality for clarity?

Reviewer #2: No further comment. Thanks to the author for clarification.

**Conclusions**

-Are the conclusions supported by the data presented?

-Are the limitations of analysis clearly described?

-Do the authors discuss how these data can be helpful to advance our understanding of the topic under study?

-Is public health relevance addressed?

Reviewer #2: No further comment. Thanks to the author for clarification.

**Editorial and Data Presentation Modifications?**

Reviewer #2: No further comment. Thanks to the author for clarification.

**Summary and General Comments**

Reviewer #1: Thank you for your warm and precise responses to each question & comment made by myself and other reviewers. I understand your sincerity and your contribution to this work.

Reviewer #2: No further comment. Thanks to the author for clarification.

PLOS authors have the option to publish the peer review history of their article (what does this mean? ). If published, this will include your full peer review and any attached files.

**Do you want your identity to be public for this peer review?** For information about this choice, including consent withdrawal, please see our Privacy Policy .

Reviewer #1: No

Reviewer #2: **Yes: ** Charls Erik Halder

Reviewer #3: No

---

## [Editor Report · Acceptance letter]

Dear Alrawashdeh,

We are delighted to inform you that your manuscript, "Temporal trends in the incidence and case severity of COVID-19 cases among the Syrian refugees in Azraq camp in Jordan: a retrospective observational study," has been formally accepted for publication in PLOS Neglected Tropical Diseases.

Best regards,

Shaden Kamhawi

co-Editor-in-Chief

Paul Brindley

co-Editor-in-Chief
